# Development of PVA Electrospun Nanofibers for Fabrication of Bacteriological Swabs

**DOI:** 10.3390/biology12111404

**Published:** 2023-11-06

**Authors:** Rebecca Pellegrino, Stefania Villani, Daniela Spagnolo, Irene Carofalo, Nico Carrino, Matteo Calcagnile, Pietro Alifano, Marta Madaghiele, Christian Demitri, Paola Nitti

**Affiliations:** 1Department of Engineering for Innovation, Campus Ecotekne, University of Salento, Via per Monteroni, 73100 Lecce, Italy; stefania.villani@unisalento.it (S.V.); nico.carrino94@gmail.com (N.C.); marta.madaghiele@unisalento.it (M.M.); christian.demitri@unisalento.it (C.D.); 2Microbiotech s.r.l., Via A. Tamborino s.n.c., 73024 Maglie, Italy; danielaspagnolo9@gmail.com (D.S.); carofaloirene@gmail.com (I.C.); 3Department of Biological and Environmental Sciences and Technologies, Campus Ecotekne, University of Salento, Via per Monteroni, 73100 Lecce, Italy; matteo.calcagnile@unisalento.it (M.C.); pietro.alifano@unisalento.it (P.A.)

**Keywords:** electrospinning, PVA, nanofibers, swab, *Pseudomonas aeruginosa*, *Staphylococcus aureus*

## Abstract

**Simple Summary:**

The enormous demand for swabs for clinical use that has characterised recent years, together with the need to develop effective, eco-friendly, and low-cost strategies for their production, has driven nanotechnological research to the development of new manufacturing processes. In this context, electrospinning represents a promising solution as it guarantees broad control in terms of fibre size and porosity and good scalability of the production process. In this study, this technique was chosen for the fabrication of biodegradable polymeric membranes to be used for the manufacture of swab tips. The electrospun mats thus obtained were characterised, and the prototype swab thus assembled was investigated in its uptake and release properties. Furthermore, the use of this prototype for the collection of bacteriological samples was preliminarily evaluated. The in vitro tests revealed a good potential, comparable to those of commercial foam swabs adopted as the reference standard and hopefully replaceable by the manufactured prototype.

**Abstract:**

In recent years, the enormous demand for swabs for clinical use has promoted their relevance and, consequently, brought the environmental issues due to their single use and lack of biodegradability to the attention of the healthcare industry. Swabs consist of a stick that facilitates their easy handling and manoeuvrability even in complex districts and an absorbent tip designed to uptake and release biological samples. In this study, we focused on the fabrication of an innovative biodegradable poly(vinyl alcohol) (PVA) nanofiber swab tip using the electrospinning technique. The innovative swab tip obtained showed comparable uptake and release capacity of protein and bacterial species (*Pseudomonas aeruginosa* and *Staphylococcus aureus*) with those of the commercial foam-type swab. In this way, the obtained swab can be attractive and suitable to fit into this panorama due to its low-cost process, easy scalability, and good uptake and release capabilities.

## 1. Introduction

Swabs are medical devices consisting of a stick and a tip, and they are generally used in several fields, from medical application [1,2] to forensics [3,4] and research [4]. The swab stick can be made of polyester, polypropylene (PP), polystyrene (PS), or ecofriendlier wood. It is important to ensure good manoeuvrability in all situations and good resistance to avoid swab breaks during its use while maintaining the needed flexibility for patients’ comfort [5]. The tip is the focus on evaluating swab effectiveness, as it is responsible for the specimen collection for cultural and molecular microbiological analyses [6]. In fact, when swabs are used in biomedical fields for the collection of microbiological specimens, the tip can influence the detection of microorganisms. Although there are many detection techniques for bacteria and viruses in biological samples, from bacterial culture [7,8] to molecular detection [9,10] up to immunological assays [11], the sample collection phase is crucial. Dacron, cotton, nylon fibres, or foam swabs require specific fabrication methods, finally resulting in different physicochemical features and efficiency in collecting microorganisms [6]. Furthermore, the nature of the swab tip must also respond to the circumstances of the collection of the biological samples: for example, intact skin, mucous membranes, or damaged wound tissues [12]. Consequently, it is often difficult to choose the right device, as the performance of the swab often varies based on the sampling target and the surface from which it was isolated [13,14]. In general, the efficiency of swab tips mostly depends on their material, which particularly influences absorption and release capacity and sample collection properties [15]. First, they were made of cotton, a biopolymer that presents high absorbency and excellent sampling collection in patient care thanks to bonds that hydroxyl groups on cellulose units can form with carbohydrates on cell membranes [16]. Unfortunately, due to their organic nature, cotton swabs may leave residues during analysis [17]. More recently, synthetic materials processed in innovative ways have mostly substituted cotton swabs in the healthcare industry [18]. In particular, foam and flocked swabs are the most used. Flocking is a technique that allows obtaining perpendicularly oriented fibres on adhesive-coated surfaces using mechanical or electro-statical techniques [19]. Flocked swabs are made in nylon fibres organised into an open fibre structure that is optimal for specimen collection. As an alternative to nylon, polyurethane (PU) can be used as swab tip material [20,21]. It is an elastomer processed in foams and sponges with a uniform open structure at its exposed surface. Despite its hydrophobic nature, its peculiar structure allows optimal absorption and release of biological specimens [22,23].

In this study, an innovative swab tip was produced using poly(vinyl alcohol) (PVA) as bulk material and electrospinning for the processing method. PVA is a synthetic polymer made from polyvinyl acetate through hydrolysis, which can be degraded by biological organisms [24,25]. It is widely used worldwide in a variety of applications, from medical [26,27,28] to food [29] fields. It was selected as a possible material for sample collection as a bacteriological swab tip because of its revealed efficiency in the recovery and detection of exhaled bacteria [30]. It has a hydrophilic nature and is soluble in water [31]. For this reason, a chemical or physical crosslinking reaction is needed to improve its stability in aqueous solutions. In this study, in order to limit the involvement of chemical reagents following the 12 Principles of Green Chemistry [32], a physical reaction by thermal crosslinking was chosen. It was proved that with thermal treatment, PVA can increase its crystallinity, resulting in an improved stability of the material [33]. To produce PVA swab tips, electrospinning was chosen as the processing method because of its simplicity and scalability and the possibility of obtaining mats constituted of nanometric fibres with an elevated surface/volume ratio that can impact on uptake capacity of the material [34,35,36,37,38]. PVA mats were characterised by their intrinsic properties and potentialities for the swab assembly through the uptake and release analysis of aqueous solutions and bacterial suspensions. In particular, *Pseudomonas aeruginosa* and *Staphylococcus aureus* were chosen in this study to evaluate the swab capacity in detecting and quantifying bacteria in a biological sample. These two human pathogenic bacteria are associated with many diseases in humans and were used to evaluate the viability of the bacteria on PVA swab tips. *P. aeruginosa* is a Gram-negative, rod-shaped, opportunistic pathogen often responsible for hospital-acquired infections in immunocompromised or lung-ill hosts, such as patients with cystic fibrosis [39]. *S. aureus* is a Gram-positive, aerobic, and spherical-shaped bacterium, often responsible for bacteraemia, infective endocarditis, and skin and prosthetic infections [40]. Both *P. aeruginosa* and *S. aureus* have been included in a group of pathogens called ESKAPE (*Enterococcus faecium*, *S. aureus*, *Klebsiella pneumoniae*, *Acinetobacter baumannii*, *P. aeruginosa*, *Enterobacter*) by the World Health Organization (WHO) as those pathogens that are resistant to most available antimicrobial agents, which limits clinical treatment options [41].

The results showed how the innovative PVA electrospun tip prototype exhibits capabilities comparable to commercial foam swabs, resulting in a promising device for the future.

## 2. Materials and Methods

### 2.1. Materials

PVA 99+% hydrolysed (Mw 89–98 kDa) and Phosphate Buffered Saline tablets (PBS) were purchased from Sigma Aldrich (St. Louis. MO, USA). All aqueous solutions were prepared with deionized water. For swab characterisation, commercial flocked and foam swabs (Microbiotech s.r.l., Maglie, Italy) were used as a standard for comparison. 

### 2.2. Electrospun PVA Mats

A 15% w/v PVA aqueous solution was prepared under magnetic stirring in a water bath at 80 °C for 3 h and kept at 45 °C overnight (ON). The homogeneous solution was poured into a 20 mL plastic syringe and kept at 50 °C to remove bubbles. The syringe was placed on a syringe pump (KDS-200-CE, KD Scientific Inc., Holliston, MA, USA) and equipped with a metallic needle of 15 mm in length and 0.8 mm inner diameter and connected to the positive electrode of a high-voltage DC power supply generator (QCHV-M40, Linari Engineering s.r.l., Pisa, Italy) for the electrospinning. The collector was a cylindrical rotating drum of 30 mm in diameter and 120 mm in length (RT-Collector Web, Linari Engineering s.r.l., Pisa, Italy) placed in front of the syringe pump, connected to the negative electrode of the high-voltage DC power supply generator, and wrapped with aluminium foil to facilitate the detachment of the electrospun membrane. The distance between the needle and the collector was set to 13 cm, with an electric field between the two electrodes of 22–25 kV. The flow rate was set to 1 mL/h, and for the collector, the rotating speed was fixed at 1000 rpm, the translation speed was 10 mm/s, the starting point of stroke was 10 mm, and the stroke length was 20 mm. The electrospinning process was carried out at room temperature (RT). PVA electrospun mats were collected, peeled off from the aluminium foil, and crosslinked under vacuum at 180 °C for 2 h in a vacuum oven (M40-VT, MPM Instruments s.r.l., Monza, Italy).

### 2.3. Swab Prototypes

A simple swab prototype was produced using PVA electrospun mats to evaluate their potentiality in this field. PVA electrospun mats were cut in rectangles, measuring roughly 10 mm wide by 20 mm length, and wrapped around the end of PS sticks, using a commercial cyanoacrylates-based adhesive (Super Attak Loctite^®^, Henkel, Milan, Italy) to fix them.

### 2.4. PVA Mats Characterisation

#### 2.4.1. Morphological Investigations

Morphological analyses were performed on PVA electrospun mats by using a Quanta FEG200 (FEI, Eindhoven, The Netherlands) scanning electron microscope (SEM). Rectangular samples (10 × 20 mm) were cut from each electrospun mat and sputter-coated with gold (thickness of ~7 nm) with an automatic sputter coater (SC500, Emscope, Hertfordshire, UK). For morphological analysis, 10 random measurements of the fibres’ diameter were taken and used to determine the average diameter. 

#### 2.4.2. Fourier Transform Infrared Spectroscopy (FT-IR)

FT-IR spectra were recorded for PVA mats before and after crosslinking at 180 °C for 2 h in Attenuated Total Reflectance (ATR) mode with a Perkin Elmer Spectrum One spectrometer (Perkin Elmer, Waltham, MA, USA) at a wavelength range of 400–4000 cm^−1^ with a resolution of 4 cm^−1^.

#### 2.4.3. Swelling

To study the swelling capacity, crosslinked PVA mats were cut in rectangles, measuring roughly 10 mm wide by 20 mm length, and immersed in 0.01 M PBS at RT for different times (15 min, 30 min, 1 h, 2 h, 5 h, and 24 h). At the set time points, samples were washed three times with distilled water, and the degree of swelling was calculated using Equation (1):(1)$$Swelling\;\left(\%\right)=\frac{w_{s}-w_{d}}{w_{d}}\times 100,$$
where *w_d_* is the weight of the sample in the dry state, and *w_s_* is the weight of the swollen sample [42]. The experiment was carried out in triplicate.

#### 2.4.4. Stability Test

To study the stability of the crosslinked PVA mats, rectangular samples (10 × 60 mm) were cut and immersed in 15 mL of 0.01 M PBS at 37 °C (Julabo GmbH, Seelbach, Germany) for different times (3, 7, 14, and 28 days). At each time point, samples were washed three times with distilled water and freeze-dried for 24 h. The percentage of weight loss was calculated using Equation (2):(2)$$Weight\;loss\;\left(\%\right)=\left|\frac{w_{f}-w_{d}}{w_{d}}\right|\times 100,$$
where *w_d_* is the weight of the sample in the dry state, and *w_f_* is the weight of the freeze-dried sample [43]. The experiment was carried out for four samples at each time point.

#### 2.4.5. Mechanical Properties

Crosslinked PVA electrospun mats were tested in their tensile mechanical properties until failure in the wet state using a universal testing machine (Zwick Roell, Ulm, Germany) equipped with a 100 N load cell at a displacement rate of 0.1 mm/s and with a preload of 0.1 N. In this case, rectangular samples of the obtained and biodegraded mats (four for each group) were cut (10 × 60 mm) and immersed in 0.01 M PBS for 2 h at RT, the time needed to reach complete hydration. Their thickness, length, and width in the wet state were measured using a digital microscope (Dino-Lite digital microscope, New Taipei City, Taiwan) equipped with a software image analysis DinoCapture 2.0. Young’s modulus (E) was calculated as the slope of the linear elastic region of the stress–strain curve at low strain values (in the range of 0–5%) [44].

### 2.5. Swab Prototypes Volume Uptake and Release

#### 2.5.1. Water and PBS Uptake

To evaluate the absorption capacity, commercial and prototype swabs were immersed in 500 µL of water or 0.01 M PBS at RT for different times (30 s, 1 min, 5 min, 15 min, and 60 min). At each time point, swab water and PBS uptakes were calculated, as in Equation (3):(3)$$Uptake\;\left(\%\right)=\frac{w_{w}-w_{d}}{w_{d}}\times 100,$$
where *w_d_* is the weight of the swab (tip + stick) in the dry state, and *w_w_* is the weight of the wet swab (tip + stick) after immersion. The experiment was carried out in triplicate.

#### 2.5.2. Bovine Serum Albumin (BSA) Uptake and Release

To evaluate the performance of the prototype swab, BSA was chosen as a model protein. An aqueous solution of BSA (1 mg/mL) was used as the uptake solution to evaluate swab adsorption. More in detail, the swab was immersed in 500 µL of the uptake solution for 1 min, at the end of which the swab was transferred into 500 µL of H_2_O and left for different times (15 min, 30 min, 60 min), thus obtaining the release solution. The Bradford Assay was used to estimate the protein amount in the uptake and release solutions [45]. First, a calibration curve was calculated starting from serial dilutions of BSA in concentrations ranging from 0.5 μg/μL to 10 μg/μL. For the protein dosage, adsorption samples were prepared as follows: 17 μL of H_2_O, 3 μL of the uptake solution, and 1 mL of the mix (Bradford reagent 1:5 diluted in H_2_O). Release samples were prepared as follows: 20 µL of the release solution and 1 mL of the mix. Each sample was analysed at 595 nm by using VIS ONDA TOUCH V-11 SCAN spectrophotometer (Giorgio Bormac s.r.l., Modena, Italy) extrapolating the unknown protein concentration from the calibration curve. The experiment was carried out in duplicate.

### 2.6. Detection of Bacteria

*P. aeruginosa* PA01 and *S. aureus* SA-1 were grown in Luria-Bertani (LB) broth with shaking at 120 rpm at 37 °C. The composition of the LB medium (per litre): 10 g of NaCl, 10 g of Tryptone, and 5 g of Yeast Extract. LB agar was supplemented with 15 g of Agar per litre. Before being used, swabs were sterilised with UV light exposure for two cycles of 5 min each, and sterility was verified by rolling on agar plates while rotating to ensure that all the surfaces of the tip contacted the agar surface. Commercial foam swabs were used as a standard for comparison. The protocols used to determine bacterial suspension uptake and release capacity refer to the M40-A2 standard “Quality Control of Microbiological Transport Systems; Approved Standard—Second Edition” [46] with some adjustments. The bacterial pre-inoculum was prepared by making a suspension in a liquid LB medium of an isolated colony selected from a fresh agar plate (24–72 h) and then incubated ON at 37 °C. The day after, bacteria were resuspended in the medium, reaching an optical density (O.D.) of 0.150 measured at 600 nm. Then, bacterial suspensions were 1:5-diluted in 0.9% NaCl solution, obtaining the bacterial inoculum (approximate 3 × 10^6^ CFU/mL) for subsequent experiments (Figure 1). 

To test their uptake capacity, electrospun prototype swabs were inoculated by placing them for 15 s in 500 µL of the bacterial inoculum, used as the control (CTRL). Swabs were placed in 1 mL of 0.9% NaCl solution, vortexed for 15 s, and 10-fold serial dilutions were prepared (t_0_ h). For specimens maintained at controlled RT, swabs were placed in 1 mL of 0.9% NaCl solution and stored for 24 h at 28 °C. At the end of the incubation period, swab tubes were vortexed for 15 s, and serial dilutions were prepared (t_24_ h). Finally, 10 µL of each dilution was spotted on an LB solid medium Petri dish and incubated at 37 °C until the next day (Figure 1a). The experiments were conducted in triplicate. 

To evaluate the swabs’ performance in rolling on agar plates, the bacterial pre-inoculum was prepared as described above and 1:5-diluted in 0.9% NaCl solution, finally reaching approximately 1 × 10^6^ CFU/mL inoculum as a starting point to prepare 10-fold serial dilutions. Then, electrospun swabs were placed in 500 µL of each dilution for 15 s and streaked over the entire sterile LB agar plate while rotating to ensure that all the surfaces of the swabs contacted the agar surface. Petri dishes were incubated at 37 °C for 24 h, and then CFUs were appreciated to qualitatively evaluate the prototype rolling performance (Figure 1b). The experiment was conducted in duplicate.

### 2.7. Statistical Analysis

The data are presented as the mean ± Standard Deviation (SD) for the indicated number of experiments. The statistical analysis was conducted by using One- and Two-way ANOVA. In all comparisons, *p* < 0.05 was considered statistically significant, and the significance level was reported when present.

## 3. Results

### 3.1. PVA Mats

To obtain innovative swab tips, PVA mats were first produced with electrospinning and characterised. The morphological analysis (Figure 2a) shows that fibres were randomly collected on the cylindrical rotating drum, with an average diameter of 325 ± 52 nm.

Since the PVA mats as obtained are soluble in water, a crosslinking reaction was induced by heating at 180 °C for 2 h. The effectiveness of this treatment was confirmed with FT-IR. In Figure 2b, the FT-IR spectra of the PVA mats as obtained and after thermal crosslinking are shown. In both cases, typical PVA absorption peaks are revealed at 3296 cm^−1^ (O–H stretching), 2938 cm^−1^ (asymmetric stretching of alkyl groups CH_2_), 2910 cm^−1^ (symmetric stretching of alkyl groups CH_2_), 1654 and 1564 cm^−1^ (water absorption), 1418 cm^−1^ (CH_2_ bending), 1330 cm^−1^ (δ (OH), rocking with CH wagging), 1141 cm^−1^ (shoulder stretching of C–O, typical of crystalline sequence of PVA), 1088 cm^−1^ (stretching of C=O and bending of OH, typical of amorphous sequence of PVA), 919 cm^−1^ (CH_2_ rocking), and 837 cm^−1^ (C–C stretching). The only difference in the two cases is the intensification of the peak at 1143 cm^−1^ (circled in green) due to the increased crystallinity of the PVA after the heating treatment, resulting in an expected increase in the stability of PVA in aqueous solutions.

A preliminary swelling test was performed to evaluate the absorption capacity of PVA mats after thermal crosslinking in aqueous solutions and the time needed to reach equilibrium in this absorption process. As shown in Figure 2c, PVA mats can absorb a high amount of 0.01 M PBS solution, with an increase in weight of more than 500% from the initial one. Fluctuations in the experimental data are due to the absorption in progress until 2 h, the time needed to reach the equilibrium. 

To evaluate the behaviour of PVA mats in aqueous solutions, a stability test was performed at different times in 0.01 M PBS solution at 37 °C. Figure 3a shows a weight loss of about 1.5% in the first 3 days of incubation, which remains mostly constant up to 28 days, at which point the weight loss is about 1.7%. As a result of the degradation that starts to occur in these times, the average diameter of the fibres calculated with SEM inspection increases up to 893 ± 44 nm at 28 days, as shown in Figure 3b and in micrographs in Figure 3c.

The tensile mechanical behaviour of PVA mats was tested, and a typical stress–strain plot obtained in these tests is reported in Figure 3d. The curve shows an initial steep rise, which indicates a high resistance to deformation of PVA mats resulting from the nanofibers’ cohesive forces due to fibre-to-fibre contacts and maximum stress at which the break occurs. The average of Young’s Modulus of crosslinked PVA mats at different times is reported in Figure 3e. Mats exhibit a great reduction in Young’s Modulus after the first three days, which reaches its lowest value at 28 days.

### 3.2. Electrospun Swab Uptake and Release Capacity

Electrospun swab prototypes were successfully produced and characterised, using commercial flocked and foam ones as standard comparisons. The SEM analysis shows the different morphologies of the three swabs compared in this study. The commercial flocked swab has both the tip and stick in nylon, a total length of 152 mm, and a breaking point of 78 mm (Microbiotech s.r.l., Maglie, Italy) and shows vertically aligned microfibres. The commercial foam swab has a PU tip, PS stick, a total length of 152 mm, and a breaking point of 78 mm (Microbiotech s.r.l., Maglie, Italy) and shows a typical rounded porosity. The electrospun prototype swab has a PVA tip, PS stick, a total length 152 mm and a breaking point of 78 mm and shows randomly oriented nanofibers (Figure 4).

Preliminary tests of water and 0.01 M PBS uptake were performed to evaluate the absorption capacity of the electrospun prototype tip compared to commercial swabs. As shown in Figure 5a, flocked swabs show a water uptake of 14% constant in time, whereas foam swabs show an increase in water uptake in the first 5 min, then it remains almost constant at 8%. Electrospun prototypes show a constant water uptake of 9% until 60 min. In 0.01 M PBS solution, performances of flocked swabs remain almost constant until 60 min, whereas in foam swabs and electrospun prototypes, PBS uptake increases until 5 min, at which time, it reaches its highest value (15% for foam swabs and 8% for electrospun prototype, respectively) (Figure 5b). A BSA solution was then used to compare commercial and prototype swab capabilities in collecting protein. Maintaining the absorption time fixed at 1 min, BSA uptake was 35% for flocked swabs, whereas foam swabs and electrospun prototypes present similar capacities with no statistically significant difference between the two (Figure 5c). BSA release is instead almost constant in time for commercial flocked and foam swabs (36% and 32%, respectively), whereas it increases over time for electrospun prototypes, reaching an average value of 40% at 60 min (Figure 5d). The statistical analysis showed no significant differences between the prototype and the foam swab, and for this reason, only this commercial swab was used as a standard comparison for the uptake and release of bacterial suspension assays. 

### 3.3. Electrospun Prototypes for the Collection of Biological Specimens

As shown in Figure 6a, the sterility of the electrospun and foam swabs after UV exposure was confirmed before being used for subsequent tests. The rolling performance was qualitatively assessed, finding similar and comparable capabilities to those found with the commercially available foam swab for both *P. aeruginosa* (Figure 6b) and *S. aureus* (Figure 6c).

The quantitative bacteriological test was used to assess the uptake and release capacity of the electrospun prototype swab compared with the commercial foam swab used as a standard for comparison (Figure 7). The bacterial uptake was set at a volume of 500 µL of bacterial suspension with a starting bacterial load of approximately 3 × 10^6^ CFU/mL for both *P. aeruginosa* and *S. aureus*. The electrospun prototype release capacity was evaluated immediately after the inoculation (t_0_ h) and after 24 h of storage at 28 °C (t_24_ h). For *P. aeruginosa* (Figure 7a), there was a nonsignificant reduction in the CFU/mL released from the swabs immediately after the inoculation, attributable to the protocol used. At the same time, bacterial overgrowth of *P. aeruginosa* (1 log10) was registered when swabs were left for 24 h at 28 °C. In any case, however, these variations are observed both in the electrospun and the foam swabs without significant differences. For *S. aureus*, there is a 1 log10 reduction in the CFU/mL released from both the prototype and the foam swabs immediately after the inoculation as compared to the starting CFU/mL. At the same time, no overgrowth phenomena were observed for *S. aureus* when swabs were stored for 24 h at 28 °C.

## 4. Discussion

The electrospinning of PVA has been widely studied for a variety of applications. In this study, the obtained electrospun fibres mats were used to fabricate an innovative absorbent tip for swabs, thanks to their nontoxicity, biocompatibility, and biodegradability. 

PVA with an elevated degree of hydrolysis was used to enhance mechanical properties and water resistance. The electrospinning process was successful, with the production of randomly oriented nanometric fibre mats without beads. Mats were then subjected to thermal crosslinking to avoid polymer dissolution in aqueous solutions. The reaction was confirmed with ATR FT-IR, where, in the detection of specific peaks of PVA [47], it was evident the increase in its crystalline peak. A swelling test confirmed that PVA mats were capable of retaining more than 500% of their weight without dissolution in the first 24 h. 

Although, for swab tips, the time needed to collect and release biological samples is short, a stability test of the material was performed to assess possible residues during analysis. The stability was confirmed for up to 28 days, at which time the weight loss was about 1.8% with a slight fibre deformation, which began to widen due to the beginning of PVA biodegradation. The PVA mats showed a plastic behaviour, with elongations greater than 30%, which allows its use for manufacturing swab tips, and a Young Modulus of 35 MPa, which decreased over time due to degradation.

To evaluate the potentialities of this material as swab tips, it was necessary to evaluate its uptake and release capacity when used as a tip for the innovative swab prototype proposed in this study, compared to commercially available models. These assays showed a constant water uptake percentage within 60 min and registered working abilities more comparable to commercial foam swabs (Figure 6a). The commercial flocked swabs showed a steady trend in PBS uptake, whereas foam swabs showed an increase in the uptake percentage until 5 min, then the absorption rate remained flat. In the case of the electrospun prototype, a constant uptake trend was observed for PBS over time, with results that are comparable to foam swabs up to 1 min (Figure 6b). Consequently, a 1-minute interval was chosen as a fixed time for BSA uptake (Figure 6c). The assessment of protein adsorption by swabs is a preliminary evaluation of their collection capacity of specimens from wounds, mucous membranes, etc., as body fluids contain a variable quantity of proteins together with water, and of bacteria, as they present appendages, such as pili, composed of proteins. In this way, good adsorption of proteins would result in an increase in specimen collection [48]. In this assay, prototype performance was comparable to foam swabs in BSA uptake from an aqueous solution. Analysing the BSA releasing capacity, electrospun prototype swabs showed an increasing trend over time, finally reaching 40% releasing capacity and better abilities than flocked and foam swabs (Figure 6d). Table 1 and Table 2 summarizes water, PBS, and BSA uptake and BSA release of electrospun prototype tips compared to commercial ones, respectively.

*P. aeruginosa* PA01 and *S. aureus* SA-1 were chosen as Gram-negative and Gram-positive bacteria, respectively, that are pathogens of the nasopharyngeal tract, to evaluate swab prototype performance in collecting bacterial suspensions. The qualitative test for the measurement of the released CFU/mL showed that the developed electrospun prototype had uptake and release capabilities of bacterial suspensions comparable to those of the commercial foam swabs chosen as a reference standard. For swabs processed immediately after inoculation (t_0_ h), there was approximately 1 log10 reduction in the CFU/mL released from the swabs if compared with the starting CFU/mL of the uptake solution, which is nonsignificant in the case of *P. aeruginosa* (Figure 7a). For swabs stored for 24 h at 28 °C, a significant 1 log10 overgrowth was observed for the Gram-negative bacterium *P. aeruginosa*. As reported in the M40-A2 Approved Standard, for specimens collected and stored before being analysed, to be considered acceptable, a decline of no more than 3 log10 in CFU/mL should be recorded between the count at t_0_ h and that at t_24_ h [46]. However, the overgrowth of some microorganisms also represents an analytical limitation in bacterial sample transport systems [46]. As expected, therefore, an overgrowth in *P. aeruginosa* was recorded but common to both electrospun and commercial foam swabs without significant differences. Table 3 summarizes the release of the bacteriological suspensions of electrospun prototype tips compared to those of commercial foam swabs.

The qualitative assay was performed to test the prototype performance in rolling on Petri dishes. The prototype was rolled after inoculation on plates containing agar medium, taking care to bring all faces of the swab tip into contact with the surface of the plate. The electrospun prototype was suitable for rolling on a plate, releasing the bacterial cells collected during the uptake without damaging the agar surface. This aspect contributes to making it suitable for possible future uses as a bacteriological swab. 

## 5. Conclusions

In this study, an innovative absorbent tip for bacteriological swabs was proposed, consisting of PVA electrospun mats wrapped around a PS stick.

PVA mats were successfully obtained through electrospinning, a processing technique that can be easily scalable for intensive manufacturing. The PVA mats obtained were stable in aqueous solutions until 28 days, with suitable mechanical properties.

In these preliminary in vitro tests, PVA mats were simply wrapped around PS sticks to simulate a swab assembly, thus evaluating material performance in this panorama. In particular, PVA electrospun tip uptake and release capacities were investigated for different kinds of samples. Firstly, it was assessed its absorption capacities using water and 0.01 M PBS solution, and then, it was simulated in a more realistic context with the detection of proteins, using BSA as a model protein, and the detection of bacteria, using *P. aeruginosa* PA01 and *S. aureus* SA-1 as Gram-negative and Gram-positive bacteria. In each test, PVA electrospun tips showed great potential, with performances comparable to foam swabs currently commercially available and used worldwide.

These results show that the use of PVA electrospun mats can be suitable for the fabrication of bacteriological swab tips, thus representing a promising starting point for future investigations.

## Figures and Tables

**Figure 1 biology-12-01404-f001:**
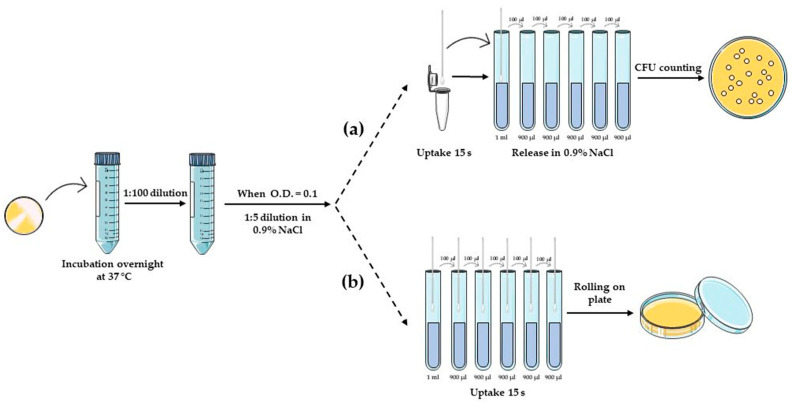
Schematic representation of the protocol used to evaluate swab uptake and release of bacterial suspensions of *P. aeruginosa* and *S. aureus*: (**a**) quantitative and (**b**) qualitative evaluation of the electrospun prototype working as a bacteriological swab.

**Figure 2 biology-12-01404-f002:**
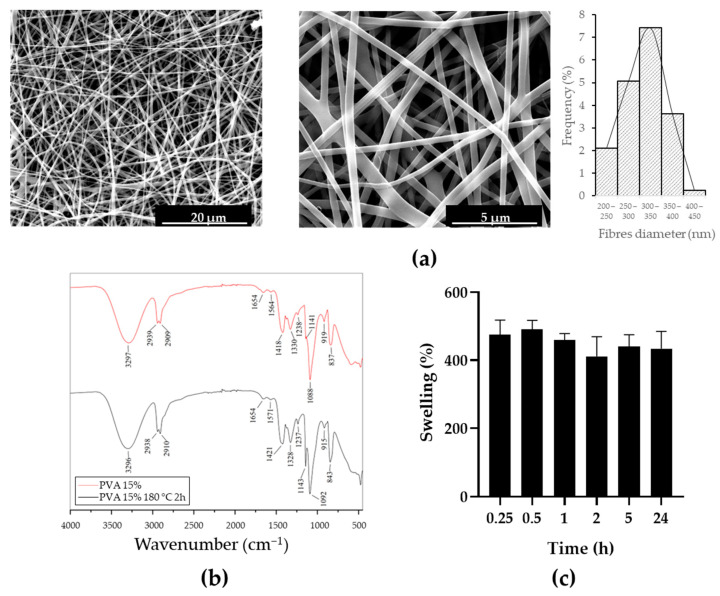
PVA mats characterisation: (**a**) SEM micrographs (5000× and 20,000×, respectively) and fibres diameter distribution; (**b**) FT-IR spectra of PVA mats before (red) and after (black) thermal crosslinking; (**c**) Swelling degree of PVA mats after thermal crosslinking in 0.01 M PBS solution.

**Figure 3 biology-12-01404-f003:**
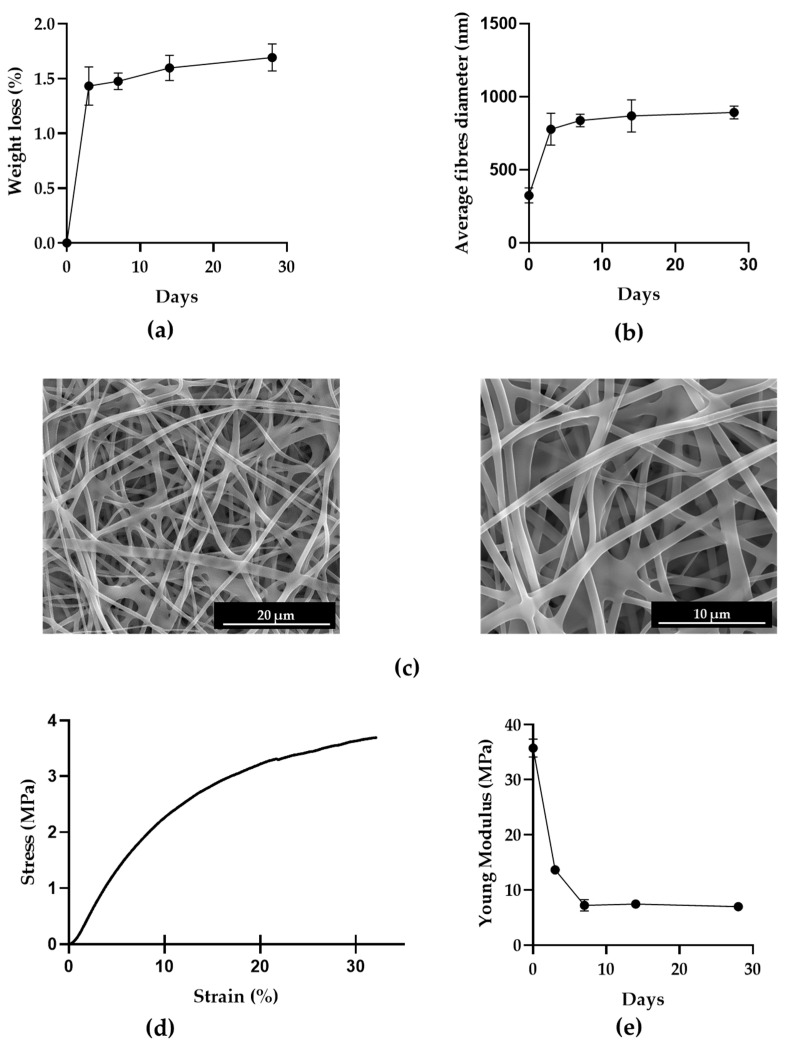
PVA mats characterisation: (**a**) Weight loss of PVA mats after thermal crosslinking in stability test; (**b**) average fibres diameter of thermally crosslinked PVA mats in stability test; (**c**) SEM micrographs (5000× and 10,000×, respectively) of PVA mats after 28 days in 0.01 M PBS at 37 °C; (**d**) example of stress–strain plot obtained in tensile test of PVA mats after thermal crosslinking; (**e**) Young’s Modulus trend of crosslinked PVA mats in time.

**Figure 4 biology-12-01404-f004:**
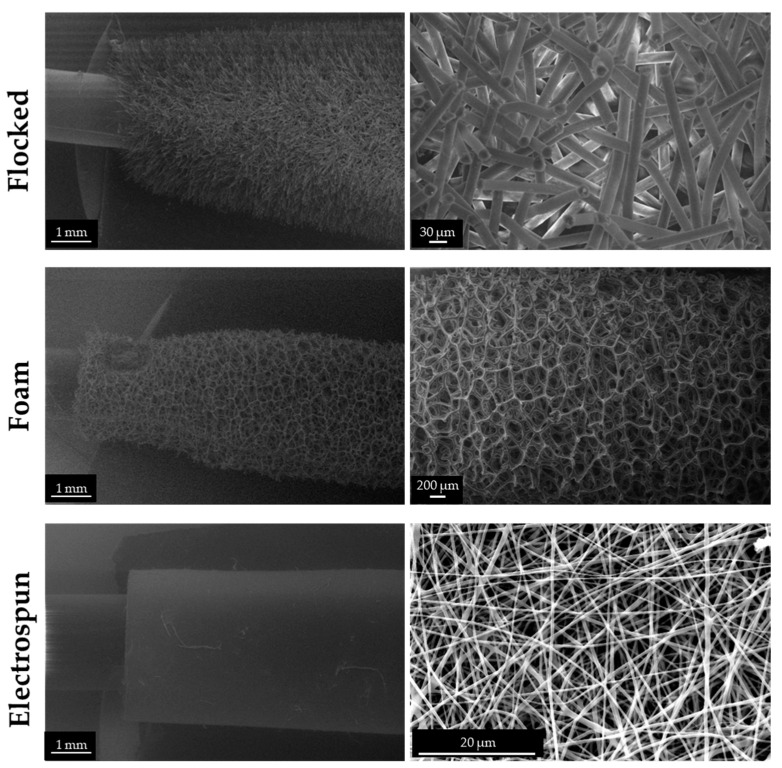
SEM micrographs of flocked (34× and 500×), foam (34× and 70×), and electrospun (34× and 5000×) swabs.

**Figure 5 biology-12-01404-f005:**
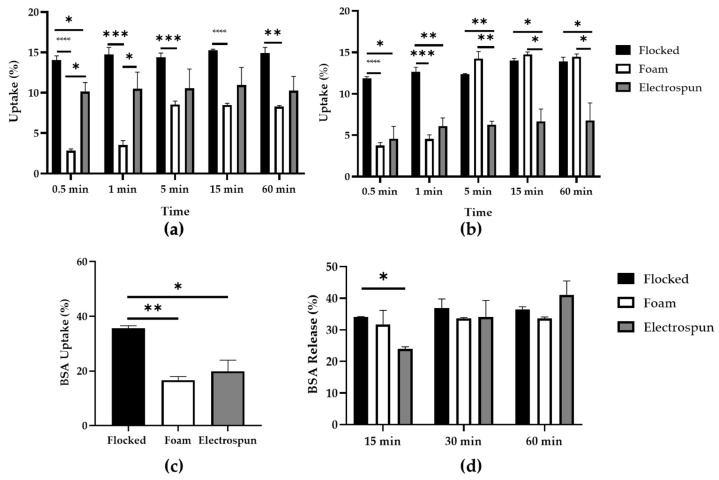
Electrospun prototype characterisation and comparison with flocked and foam swabs: (**a**) water uptake; (**b**) 0.01 M PBS uptake; (**c**) BSA uptake; (**d**) BSA release. * *p* < 0.05; ** *p* < 0.01; *** *p* < 0.001; **** *p* < 0.0001.

**Figure 6 biology-12-01404-f006:**
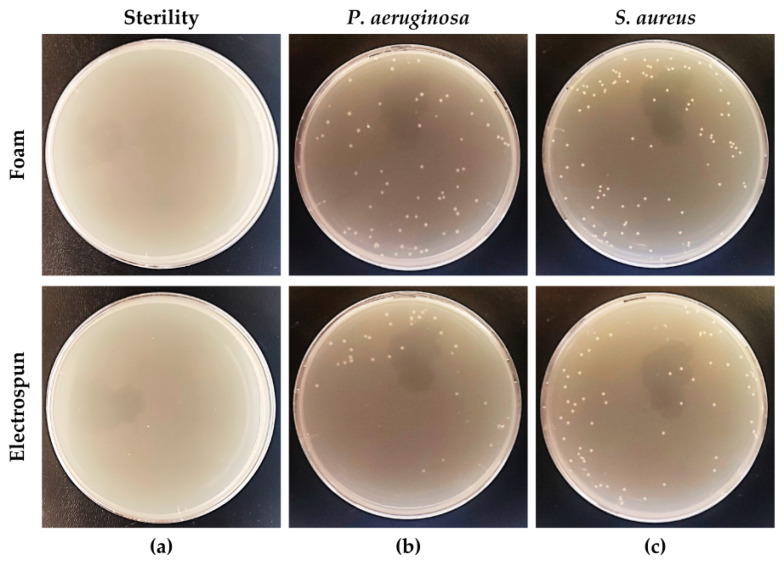
Evaluation of electrospun prototypes as bacteriological swabs: (**a**) assessment of sterility; qualitative evaluation of plate-rolling capabilities after inoculation with (**b**) *P. aeruginosa* or (**c**) *S. aureus*.

**Figure 7 biology-12-01404-f007:**
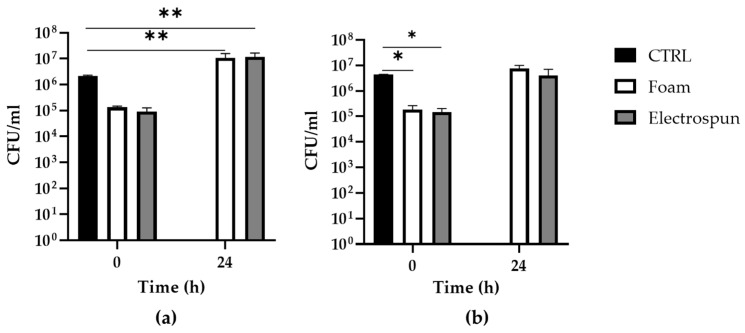
Electrospun prototype characterisation as bacteriological swabs for the uptake and release of bacterial suspensions immediately after inoculation with bacterial inoculum (used as a control, CTRL) and after 24 h of incubation at 28 °C: (**a**) *P. aeruginosa*; (**b**) *S. aureus*. * *p* < 0.05; ** *p* < 0.01.

**Table 1 biology-12-01404-t001:** Water, PBS, and BSA uptake of flocked, foam, and electrospun tips.

Swabs Uptake (%)			
	Flocked Swab	Foam Swab	Electrospun Swab
Water	14.73 ± 0.90	3.54 ± 0.54	10.52 ± 2.05
PBS	12.62 ± 0.57	4.54 ± 0.51	6.08 ± 0.99
BSA	35.57 ± 1.01	19.68 ± 0.15	19.92 ± 3.99

**Table 2 biology-12-01404-t002:** BSA release of flocked, foam, and electrospun tips.

Swabs Release (%)									
	Flocked Swab			Foam Swab			Electrospun Swab		
	15 min	30 min	60 min	15 min	30 min	60 min	15 min	30 min	60 min
BSA	34.10 ± 0.11	36.88 ± 2.91	36.48 ± 0.84	31.75 ± 4.43	33.57 ± 0.31	33.71 ± 0.37	23.99 ± 0.66	34.06 ± 5.24	41.10 ± 4.37

**Table 3 biology-12-01404-t003:** Bacteriological suspension release of foam and electrospun tips.

Bacteria Detection (%)				
	Foam Swab		Electrospun Swab	
	T_0_	T_24_	T_0_	T_24_
*P. aeruginosa*	1.3 × 10^5^ ± 0.2 × 10^5^	1.1 × 10^7^ ± 0.5 × 10^7^	0.9 × 10^5^ ± 0.4 × 10^5^	1.2 × 10^7^ ± 0.5 × 10^7^
*S. aureus*	1.9 × 10^5^ ± 0.8 × 10^5^	7.5 × 10^6^ ± 2.5 × 10^6^	1.5 × 10^5^ ± 0.6 × 10^5^	4.0 × 10^6^ ± 3.1 × 10^6^

## Data Availability

Data are contained within the article.

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
