# Peer review of "Development of PVA Electrospun Nanofibers for Fabrication of Bacteriological Swabs"

_biology, 2023, doi:10.3390/biology12111404_

Round 1

Reviewer 1 Report

Comments and Suggestions for Authors

Dear Authors,

I believe the manuscript entitled "Development of PVA electrospun nanofibers for fabrication of 2 bacteriological swabs" should be accepted for publication in its current form and after a minor revision and some additional supplemental material that I would like to see.

This paper describes a well-designed and adequately conducted set of experiments. The methodology and the follow-up analytical tools applied are sufficient. They confirm the hypothesis and main discussion topics of the paper. The quality of English is excellent. The grammar, usage, and overall readability of the manuscript are good. The technical composition of the writing, figures and graphs is also quite good and understandable. The scientific message is clear and well-understood to the reader. The topic is attractive with high potential for commercialization since the methodology and results employed for producing described newly synthesized swabs seem cost-effective, safe, eco-friendly, and biodegradable. The scientific background is well set for these results to emerge in general importance and usefulness of innovative absorbent PVA-based tips for bacteriological swabs. The selected technique is efficient and robust, allowing intensive manufacturing, thus attributing to the general impact of the paper. 

The minor revision includes the following: if possible, I would only suggest improving the quality of the graphs presented in Figure 3. I would also like to see the representation of tables with rough data from which the results presented in graphs are derived.

To conclude, I believe this paper can be published in its current form with a proposed minor revision.

Additional Comments:

1. The main aim of this research is to provide suitable and efficient material for swab tips obtained by robust and eco-friendly methods, which are at the same time cost-effective. Regarding pure scientific contribution, it is not spectacular. However, it can be considered a novelty since the authors are the first, at least according to my knowledge, who conducted such experiments and gained a product with the proposed characteristics.

2. The gaps in the field of diagnostics are always open. The improvements are welcomed for many reasons: one of them is the increased number of infections and, consequently, more residues, which can be considered detrimental material aimed to be appropriately destroyed. If the authors synthesized PVA-based swab tips, which are biodegradable, it would be helpful in general. There is always a place for new products if their application can be useful without printing to the environment.

3. The research presented in this manuscript is very concise and coherent. The uniqueness is the procedure by which the tips are manufactured. I believe this is the research's main value: the selection of material (PVA) and methods (electrospinning).

4. The methodology is simple, correct, and sufficient. The nano-characterization of the morphological representation of material is conducted with Scanning Electron Microscopy; the tests for material stability, mechanical properties, stability, swelling, and interaction with proteins (investigated interaction with BSA) are standard methods used for this purpose since they imitate physiological conditions. Also, tests conducted for detecting selected bacteria species are within the frame of referential topics. This is a preliminary study, so I believe the represented material is sufficient. The only question is if the proposed material meets the Biology Journal criteria.

5. The conclusions derived from the results and discussion are correct and comparable with referent materials already in commercial use. I understand this study as preliminary and believe that additional in vivo tests are needed for further steps towards commercialisation.

6. The references are appropriate and in line with the comments of the authors.

7. I propose the improvement of Figures containing graphical representation, and I would also like to see the rough data from which the graphs are derived form. My assumption is to believe the authors that the data is original. From the results represented in the manuscript entitled "Development of PVA electrospun nanofibers for fabrication of 2 bacteriological swabs", I could only conclude that the goal was achieved: the PVA swab tips are newly synthesized with suitable, efficient and robust methods, and they initially express the properties for which they are aimed.

Respectfully,

Reviewer

Reviewer 2 Report

Comments and Suggestions for Authors

Dear Editor

I would like to thank you for the devoting this paper to me for revision.

This paper explains the production of PVA  electrospun nanofibers as a bacteriological swab. The paper needs to some revisions to publish in Biology.

PVA is mostly used in nanofiber production. This is not novel. Authors should explain better why PVA is used.

Recently, similar works based on different polymers have been employed for the fabrication of electrospun nanofibers to test bacterial adhesion. What are the advantages and disadvantages of this work?

Please add histograms for the distribution of nanofibers’ dimater

Please SEM micrographs after the inbucation of nanofibers with water. Because PVA is hyrdophylic polymer so it can be soluble after water teratment if the crosslinking is not enough.

I believe that a table comparing the performances of the nanofibers in this study with the existing swab materials  is missing

I think allergic tests should be done on the samples.

Reviewer 3 Report

Comments and Suggestions for Authors

1.     For all equations, a times sign should be used instead of the letter ‘x’.

2.     It is recommended to check the font size requirements of the figures. Some words in Figure 1 are too small to read. The font size in Figure 2 is not consistent.

3.     The scale bars in Figure 2a and Figure 4 (electrospun) are not clear.

4.     Swabs are designed to uptake samples in a very short time (usually less than 1 min). Under this condition,    why do the authors conduct the swelling tests for up to 24 hours, weight loss, and average fiber diameter for up to 28 days? Again, the swabs are not used for such a long time.

5.     What are the dimensions of the samples for mechanical testing? A schematic image should be provided.

6.     How did the authors read the stress value from the mechanical testing instrument? The samples used in this study are not standard samples. The readout should be usually force and displacement.

7.     The color legend for Figure 5 is not distinguishable. It is suggested to use different infill patterns.

8.     The uptake measurement for water and PBS is conducted under different time conditions. Why didn’t the BSA uptake measurement show the same results as the other two?

9.     Similar to Comment 4, why do the authors conduct BSA release measurements for up to 1 hour?

Comments on the Quality of English Language

1.     Line 29: ‘the healthcare industry their relevance’. There is a need for transition words between ‘industry’ and ‘their relevance’.

2.     Line 246: ‘diameter of (353 ± 58) nm’  ‘diameter of 353 ± 58 nm’

3.     Line 418: ‘it was proposed an innovative absorbent tip for bacteriological swabs,’  ‘innovative absorbent tip for bacteriological swabs was proposed,’
